

# Morphological features and water solubility of iron in aged fine aerosol particles over the Indian Ocean

Sayako Ueda[1], Yoko Iwamoto[2], Fumikazu Taketani[3], Mingxu Liu[1], Hitoshi Matsui[1]

[1] Graduate School of Environmental Studies, Nagoya University, Nagoya, 464-8601, Japan

[2] Graduate School of Integrated Sciences for Life, Hiroshima University, Hiroshima University, Higashi-Hiroshima, 739-8521, Japan

[3] Japan Agency for Marine-Earth Science and Technology, Yokohama, 237-0061, Japan

*Correspondence to*: Sayako Ueda (ueda.sayako.u2@f.mail.nagoya-u.ac.jp)





**Abstract.** Atmospheric transport of iron (Fe) in fine anthropogenic aerosol particles is an important route of soluble Fe supply to remote oceans from continental areas. To investigate Fe properties of aerosol particles over remote oceans, we analyzed atmospheric aerosol particles over the Indian Ocean during the research vessel Hakuho Maru KH-18-6 cruise. Aerosol particles collected using a cascade impactor were analyzed using

transmission electron microscopy (TEM) with an energy-dispersive X-ray spectrometry analyzer. The particle shape and composition on the sample stage of 0.3–0.8 μm aerodynamic diameter indicated that most particles collected north of the equator were composed mainly of ammonium sulfate. Regarding the particle number fraction, 0.6–3.0% of particles contained Fe, which mostly co-existed with sulfate. Of those particles, Fe was found 26% as metal spheres, often co-existing with Al or Si, regarded as fly ash, 14% as mineral dust, and 7%

as iron oxide aggregations. Water-dialysis analyses of TEM samples indicated that Fe in spherical fly ash was almost entirely insoluble, whereas Fe in the other morphological-typed particles was partly (65% Fe mass on average) soluble. Global model simulations mostly reproduce observed Fe mass concentrations in $PM_{2.5}$ collected using a high-volume air sampler, including their north–south contrast during the cruise. In contrast, a marked difference was found between the simulated mass fractions of Fe mineral sources and the observed Fe

types. For example, the model underestimated anthropogenic aluminosilicate Fe contained in matter such as fly ash from coal combustion. Our observations suggest that Fe in particles over remote ocean areas has multiple shapes and minerals, and further suggest that its solubility after aging processes differs depending on their morphological and mineral type. Proper consideration of such Fe types at their sources is necessary for accurate estimation of atmospheric Fe effects on marine biological activity.



## 1 Introduction

Iron (Fe) is recognized as an essential micronutrient for ocean primary productivity. Addition of water-soluble
Fe to remote oceans, designated as a high-nutrient and low-chlorophyll region, stimulates phytoplankton blooms.
Iron can thereby change the marine environment and biological diversity, and the global carbon cycle (Martine
and Fitzwater, 1988; Baar et al., 1995; Harrison et al., 1999; Jickells et al., 2005; Tsuda et al., 2003 and 2007;
Iwamoto et al., 2009). Transport and deposition of atmospheric aerosol particles is an important route supplying
Fe to remote ocean areas. Main sources of soluble Fe in the atmosphere are Fe-containing mineral dust and Fe
emitted from anthropogenic combustion and biomass burning (combustion Fe) (e.g. Guieu et al., 2005;
Mahowald et al., 2009). Mineral dust is emitted mainly as coarse particles by dust storms from arid and semi-
arid continental areas. By contrast, combustion Fe is emitted as both fine and coarse particles, through
evaporation and condensation processes (Markowski and Filby, 1985; Liu et al., 2018; Ohata et al., 2018).
Although many earlier studies have implicated mineral dust as an important source of soluble Fe to oceans (e.g.
Uematsu et al., 1893; Mahowald et al., 2005; Iwamoto et al., 2011), anthropogenic Fe has also attracted
increasing attention recently as a source supplying water-soluble Fe, steadily and efficiently, by long-range
transport (Chuang et al., 2005; Sedwick et al., 2007; Luo et al., 2008; Takahashi et al., 2013; Ito, 2015; Matsui
et al., 2018b).

Recent sophisticated global aerosol modeling studies have evaluated global climatic effects of anthropogenic
Fe (e.g., Scanza et al., 2018; Matsui et al., 2018b; Rathod et al., 2020). Matsui et al. (2018b) demonstrated that
the atmospheric burden of anthropogenic combustion Fe is eight times greater than earlier estimates. Simulation
using a soluble Fe mechanism designed for Earth system models by Scanza et al. (2018) incorporates
consideration of changes of Fe solubility with atmospheric processing of Fe in dust and combustion aerosols.
They concluded that, in many remote ocean regions, sources of Fe from combustion and dust aerosols are
equally important. Moreover, Rathod et al. (2020) released a revised emission inventory of anthropogenic
combustion Fe using a technology-based methodology. However, the accuracy of current model-based estimates



remains unclear because of the lack of information related to the mineral composition, morphological structure, and solubility of actual Fe-containing particles in the atmosphere, especially in remote ocean areas.

Actually, Fe is emitted as aerosol particles having various morphologies, with their mineralogy and size
distributions according to their sources. Particle size and composition are related to the particle lifetime and water-solubility of Fe. In addition, changes of Fe solubility with particle aging processes depend on many factors such as Fe mineralogy and size, atmospheric and meteorological conditions, and particle acidity (Wiederhold et al., 2006; Journet et al., 2008; Cwiertny et al., 2008; Shi et al., 2009 and 2015; Ito and Feng, 2010; Li et al., 2017; Sakata et al., 2022). Earlier knowledge about relations between solubility, Fe mineral species, and aging
process has usually been based on bulk sample measurements, laboratory experiments, and simulations. However, information about individual Fe-containing particles that have experienced actual atmospheric transport remains insufficient. To evaluate model results, Fe mass concentrations and solubilities measured during numerous observation studies and chemical analyses of bulk samples have been used (Mahowald et al., 2009; Wang et al., 2015; Myriokefalitakis et al., 2018; Rathod et al. 2020). Nevertheless, data of bulk samples
alone do not provide information about the source, mineralogy, atmospheric aging, or solubility of individual Fe-containing particles. Enhancing our understanding of the roles and effects of atmospheric Fe on marine environments necessitates the elucidation of details of atmospheric Fe properties in remote areas far from their sources.

Compared to major aerosols such as sea-salt and sulfates, Fe-containing particles constitute a minority in remote
areas. This relative scarcity makes it difficult to find and investigate Fe-containing particles in aerosol samples. However, some observation studies conducted in leeward areas of polluted regions have revealed trace metals in individual particles (Hidemori et al., 2014; Li et al., 2017). For example, Li et al. (2017) investigated individual Fe-containing particles aged for 1–2 days using single-particle analysis of samples collected under polluted air over the East China Sea. Using scanning transmission electron microscopy (STEM) and nanoscale
secondary ion mass spectrometry, they demonstrated the presence of iron sulfate in a sulfate coating around



iron oxide ($FeO_x$) as evidence of Fe aging. Sample collection leeward of polluted regions and recent microscopic techniques has made it possible to find minor Fe in aged particles. As a technique of microscopic analysis, water dialysis is a powerful tool for the investigation of the ratio of water soluble and insoluble materials in individual particles (Okada et al., 1983; Miki et al., 2014; Ueda et al., 2011ab, 2018, 2022). This method comprises morphological observations and comparison before and after water dialysis of aerosols. Combination with energy-dispersive X ray spectrometry (EDS) can quantify water-soluble elements in individual particles (Ueda et al., 2022).

For this study, we conducted sampling to investigate aged Fe-containing particles onboard the R/V Hakuho Maru KH-18-6 cruise of November 2018 over the Indian Ocean. South Asian regions have severe air pollution problems even now, attributable to anthropogenic and natural sources (Gyttukunda et al., 2014; Chen et al., 2020; Dhaka et al., 2020; Kanawade et al., 2020; Ojha et al., 2020; Takigawa et al., 2020). Sea areas around the KH-18-6 cruise shipping route are not areas with Fe limitations on primary production by marine microorganisms (Mahowald et al., 2018), but are suitable for catching aerosols in long-range transported polluted air from South Asia. By using transmission electron microscopy (TEM) analysis with EDS and water dialysis in this study specifically examining samples of fine particles that can contain combustion aerosols, we investigated morphological features of Fe-containing particles related to the origin and atmospheric aging process. As basic data for aerosols, bulk aerosol sampling of $PM_{2.5}$ was conducted to measure ions and metals. After describing the methods (Sect. 2), this paper presents the mass concentration and number fraction of Fe and their relation with other major aerosol components internally and externally co-existing with Fe, based on bulk samples and TEM observations (Sects. 3.1 and 3.2). Then, typical morphological features of Fe-containing particles are explained in terms of their relation to the Fe source (Sect. 3.3). Additionally, this report describes global model simulations of each source Fe and comparison with results obtained from observations (Sect. 3.4). Finally, differences of measured solubility for morphologically categorized Fe are presented with discussion of the relation with atmospheric aging and implications from these simulations of Fe (Sect. 3.5).





## 2 Methods

### 2.1 Atmospheric observations on board and air mass backward trajectories

Atmospheric observations were conducted over the Indian Ocean during the R/V Hakuho Maru during KH-18-6, which took place on 6–28 November 2018. Figure 1 portrays ship tracks of the R/V Hakuho Maru cruise, 5-day backward air mass trajectories, and sampling locations of TEM samples. The backward trajectories were computed using the Hybrid Single-Particle Lagrangian Integrated Trajectory (HYSPLIT) model developed by the National Oceanic and Atmospheric Administration (NOAA) Air Resources Laboratory (ARL) (Stein et al., 2015; Rolph et al., 2017). The settings of the trajectory duration, starting height, vertical mode calculation method, and dataset were chosen, respectively, as 5 days, 500 m above sea level, model vertical velocity, and GDAS meteorological data. Air masses of the northern Indian Ocean (6–16 November 2018) derived from India. Those around the equator were from the east (6–19 November 2018); those of the southern Indian Ocean were from the sea around western Australia, moving counterclockwise to the observation sites (20–28 November 2018).

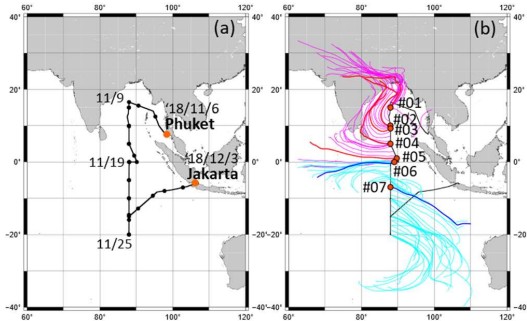

**Figure 1: Ship tracks of KH-18-06 cruse of the R/V Hakuho Maru (a) and 5-day horizontal backward air mass trajectory (b). Black dots of (a) represent 0:00 am of each day at universal time. Calculations of backward air mass trajectory started from 500 m a.s.l., above the site. Magenta and cyan thin lines respectively show trajectories of every 6 h for sites north and south of the equator. Orange circles of (b) represent TEM sampling sites. Red and blue lines show trajectories for TEM sampling.**



## 2.2 Chemical composition of $PM_{2.5}$

We collected $PM_{2.5}$ samples on Teflon filters (WP500-50; Sumitomo Electric Fine Polymer, Inc.) and prebaked

(900 °C for 3 h) quartz fiber filters (QR-100; Advantec Toyo Kaisha Ltd.) using two high-volume samplers (HV-700F; Shibata Science Co. Ltd.) with a custom-made particle-size separator at about 12-h or 24-h intervals at a flow rate of 500 L min$^{-1}$ on the compass deck (a.s.l. 14 m). To avoid collecting particles from the ship exhaust (the ship's funnel was located to the rear of the sampling position), the pumping of aerosol samplers was controlled automatically using a wind sector to operate only when the relative wind direction was −80° to

80° of the bow and relative wind speed was higher than 3 m s$^{-1}$. After collection, the Teflon and quartz fiber filter samples were stored respectively at 4 °C and −18 °C before chemical analyses (for ionic species and trace metals) at the home laboratory. Mass concentrations of water-soluble ionic species ($Cl^-$, $NO_3^-$, $SO_4^{2-}$, $Na^+$, $NH^+$, $K^+$, $Mg^{2+}$, and $Ca^+$) on quartz fiber filter samples were analyzed using ion chromatography. Non-sea-salt (nss) concentrations of $SO_4^{2-}$, $K^+$, and $Ca^+$ were estimated from $Na^+$ concentrations in the samples using the bulk

seawater ratios described by Wilson (1975). Metals in $PM_{2.5}$ (Na, Al, K, Ca, Ti, V, Mn, Fe, Ni, and Zn) were analyzed using inductively coupled plasma mass spectrometry (ICP-MS, 7700X, G3281A; Agilent Technologies, Inc.) with microwave-assisted extraction in a mixture of nitric acid, hydrofluoric acid, and hydrogen peroxide using the Teflon filter sample.

## 2.3 Individual particle analyses using an electron microscope

### 2.3.1 Observation and elemental analysis

Aerosol particles were collected for morphological particle analysis using TEM. Aerosols after diffusion drying were collected on carbon-coated nitrocellulose (collodion) films using cascade impactors. The 50% cut-off diameters of the three stages 1, 2, and 3 were, respectively, 1.6 μm, 0.8 μm, and 0.3 μm. Aerosol samples were collected at the upwind side on the compass deck of the ship for 10–30 min at a flow rate of 1.0 L min$^{-1}$. The



TEM samples were taken at about 1–2 samples per day and were stored under dry conditions at room temperature (about 25°C) until TEM analyses were conducted at Nagoya University. For this study, seven samples (#01–07) of stage 3 were used for analyses. Sample collection sites and the 5-day backward trajectories are depicted in Fig. 1b. Sample details are presented in Table 1.

**Table 1:  TEM samples used for this study and analyzed particle and Fe-containing particle numbers**

| Sample ID | Sampling time | | Location | | Atmospheric conditions | | EDS analyzed particles | Fe-containing particles | |
|---|---|---|---|---|---|---|---|---|---|
| | start time | period | Lat. | Long. | Temp. | RH | | | |
| | YYYY/MM/DD h:mm | min | N deg. | E deg. | °C | % | number | number | % |
| #01 | 2018/11/10 2:43 | 10 | 14.99 | 87.99 | 28.8 | 66 | 535 | 3 | 0.6 |
| #02 | 2018/11/12 1:51 | 10 | 10.02 | 87.98 | 28.9 | 81 | 305 | 2 | 0.7 |
| #03 | 2018/11/13 2:01 | 20 | 9.24 | 88.00 | 28.8 | 80 | 703 | 17 | 2.4 |
| #04 | 2018/11/14 2:36 | 20 | 5.01 | 87.98 | 28.8 | 74 | 507 | 10 | 2.0 |
| #05 | 2018/11/16 2:31 | 20 | 1.01 | 89.72 | 29.5 | 69 | 336 | 10 | 3.0 |
| #06 | 2018/11/18 8:05 | 25 | -0.01 | 89.06 | 32.3 | 60 | 136 | 2 | 1.5 |
| #07 | 2018/11/21 7:56 | 31 | -6.92 | 88.00 | 29.3 | 71 | 106 | 1 | 0.9 |


Particles collected on the collodion film were photographed using TEM (200 keV, JEM-2100 plus; JEOL Ltd.) at 1.2 k and 6 k × magnifications. To measure the heights of individual particles on the collection surface, particles were coated with a Pt/Pd alloy at a shadowing angle of 26.6° (arctan 0.5) before being micrographed.

The Pt/Pd coating thickness was about 7 Å. The EDS analyses were conducted using the TEM operated in

STEM mode at 200 keV. Similar analyses of TEM and EDS hyperspectral imaging (HSI) data were described by Ueda et al. (2020 and 2022) and by Ueda (2021). For this study, EDS-HSI data were sampled at greater than 20 k magnification for 10–30 frames (20 s per frame) and were kept for each dot of 256 × 256 pixels using software (NSS3; Thermo Fisher Scientific Inc., Hampton, NH, USA). The dot size was 26 × 26 nm in the

observed field at 20 k magnification. Although estimation of the mass of each element from EDS analysis is difficult, software can estimate the mass fraction ($MF_X$) of an element $X$ to all analyzed elements from measurement results of X-ray counts as relative values to detected elements. Elemental analyses were conducted fundamentally for C, N, O, Na, Mg, Al, Si, P, S, Cl, K, Ca, Ti, V, Mn, Fe, Cu, Zn, Pd, and Pt. Although the software can identify another element automatically if EDS spectra have a specific X-ray peak, other elements

were detected only rarely. To obtain the elemental compositions of individual particles, the $MF_X$ of each element was obtained manually for selected areas according to the particle shape and size of each. Noise effects of the background were eliminated using mass normalized by the Pd mass ($X/Pd$) estimated as a ratio of $MF_X$ to mass fraction of Pd ($MF_{Pd}$). Actually, Pd vapor-deposited to the whole top surface of samples at the shadowing method is not included in general aerosols. In addition, samples #01–07 were vapor-deposited by Pt/Pd at one

time. The $X/Pd$ can be treated as an independent value from the other components of aerosol, collodion film thickness, and distance from the Cu grid. If the difference of the $X/Pd$ value of a particle area to that of a near background area was greater than three standard deviations of multiple background spectra in the same sample, then the value was treated as a significant spectrum of the particle.

**2.3.2 Water dialysis and estimation of water-soluble Fe fraction ($f_{WSFe}$)**

For samples collected from the Indian Ocean (samples #01–#05) north of the equator, a water dialysis technique (Mossop, 1963; Okada, 1983; Okada et al., 2001; Ueda et al., 2011ab, 2018, 2022) was applied to investigate the mixing states of water soluble and insoluble materials in particles. The TEM grid with particle samples was floated on ultrapure water at about 25°C for 3 h with the collection side upward. After water dialysis, some



areas were photographed again. However, a large part of the collodion film unfortunately tore during water

dialysis. For sample #03 only, some EDS analysis data for Fe-containing particles were obtained from the same

particle area after water dialysis.

Using the Fe mass normalized by Pd mass (*Fe/Pd*) before and after water dialysis of sample #03, the fraction

of water-soluble Fe to total Fe in individual Fe-containing particles, $f_{\text{WSFe}}$, was estimated. The fraction of water-

soluble Fe to total Fe in a particle, $f_{\text{WSFe}}$ can be represented as


$$f_{\text{WSFe}} = \left(1 - \frac{m_{\text{WIFe}}}{m_{\text{Fe}}}\right) \times 100 \ [\%], \tag{1}$$

where $m_{\text{WIFe}}$ and $m_{\text{Fe}}$ respectively denote the masses of water-insoluble Fe and total Fe. For this study, these

values can be represented respectively as Fe masses after and before water dialysis. Because vapor-deposited

Pd is water insoluble, the mass of Pd after water dialysis can be regarded as unchanged from that before water

dialysis. In this case, the ratio of $m_{\text{WIFe}}$ and $m_{\text{Fe}}$ of a Fe-containing particle can also be replaced by the ratio of

*Fe/Pd* before and after water dialysis for the same region (*Fe/Pd*$_{\text{before}}$ and *Fe/Pd*$_{\text{after}}$ , respectively) as

$$f_{\text{WSFe}} = \left(1 - \frac{Fe/Pd_{\text{after}}}{Fe/Pd_{\text{before}}}\right) \times 100 \ [\%] , \tag{2}$$


which comprises only values that were measurable using EDS analysis in this study. Details of this estimation

method are explained in Supplement S1.

**2.4 Global model simulation of Fe**

We conducted global model simulations using the Community Atmosphere Model (ver. 5; CAM5) with the

Aerosol Two-dimensional bin module for foRmation and Aging Simulation (ver. 2; CAM5-chem/ATRAS2),



with modifications for particulate iron (Matsui et al., 2014, 2018a; Matsui and Mahowald, 2017; Matsui, 2017; Liu and Matsui, 2021a, 2021b; Liu et al., 2022). Model setting for this study was described by Liu et al. (2022). Briefly, the model incorporates emissions, gas-phase chemistry, condensation or evaporation of inorganic and organic species, coagulation, nucleation, activation of aerosols and evaporation from clouds, aerosol formation

in clouds, dry and wet deposition, aerosol optical properties, aerosol–radiation interactions, and aerosol–cloud interactions. Aerosol particles were resolved with 12 size bins from 0.001 to 10 μm dry diameter. The model was run with horizontal resolution of 1.9° × 2.5° and 30 vertical layers from the surface to approximately 40 km. The near-surface layer of model results was used for this study.

The model explicitly treats Fe from biomass burning and anthropogenic combustion. Five Fe sources/minerals

(biomass burning, and four anthropogenic Fe (magnetite ($Fe_3O_4$), hematite ($Fe_2O_3$), kaolinite ($Al_2Si_2O_5(OH)_4$) and illite ($(K, H_3O)(Al,Mg,Fe)_2(Si, Al)_4O_{10}$)) are considered in addition to eight other aerosol species (sulfate, nitrate, ammonium, dust, sea salt, primary and secondary organic aerosol, BC, and water). The anthropogenic Fe emission inventory developed by Rathod et al. (2020) with the update by Liu et al. (2022) for Southern Africa was used to model global-scale atmospheric iron concentrations. The size distribution of anthropogenic Fe was

referred from observation results reported by Moteki et al. (2017) for magnetite over eastern Asia from aircraft measurements using a single-particle soot photometer. Combustion iron emissions from open biomass burning were calculated based on Luo et al. (2008). Dust Fe was not treated in our model, but we assumed a constant iron content of 3.5% in natural dust (Duce et al., 1991; Jickells et al., 2005; Shi et al., 2012).

## 3 Results and Discussion

### 220 3.1 Horizontal variation of PM$_{2.5}$ components

Figure 2 shows horizontal distributions of nss-$SO_4^{2-}$, $NH_4^+$, and Fe in PM$_{2.5}$. Average, 25th percentile, and 75th percentile mass concentrations of ions and metals north and south of the equator at 87–90°E are presented



respectively in Tables 2 and 3. Scatter plots showing concentrations of Fe and the other elements are supported in Figs. S1 and S2 of Supplemental Materials. Among the measured ions in PM$_{2.5}$, the sum of the mass fractions

of nss-SO$_4^{2-}$, NH$_4^+$, Na$^+$, and Cl$^-$ was larger than 84% of total ion mass concentrations. North of the equator, nss-SO$_4^{2-}$ and NH$_4^+$ concentrations were especially high. These mass fractions were, respectively, 70–76% and 18–22%, except for data of 11 November 2018 when rain events occurred. The values of nss-K$^+$, which are regarded as originating mainly from biomass burning (Andreae, 1983; Kawamura and Kaplan, 1987; Narukawa et al., 1999), also tended to be north of the equator. South of the equator, the nss-SO$_4^{2-}$ and NH$_4^+$ concentrations

were lower. Consequently, the fractions of sea salt components (i.e. Na$^+$ and Cl$^-$) increased. The PM$_{2.5}$ ionic amount in equivalent concentrations of total cation without H$^+$ were comparable to or greater than 75% of that of total anion (Fig. S1a). For non-sea-salt components, the relations between the doubled nss-SO$_4^{2-}$ molar concentration and the NH$_4^+$ plus nss-K$^+$ molar concentration were usually between 1:1 and 2:1 (Fig. S1b), suggesting that nss-SO$_4^{2-}$ originated from ammonium sulfate, ammonium bisulfate, and ammonium potassium

rather than from sulfuric acid.

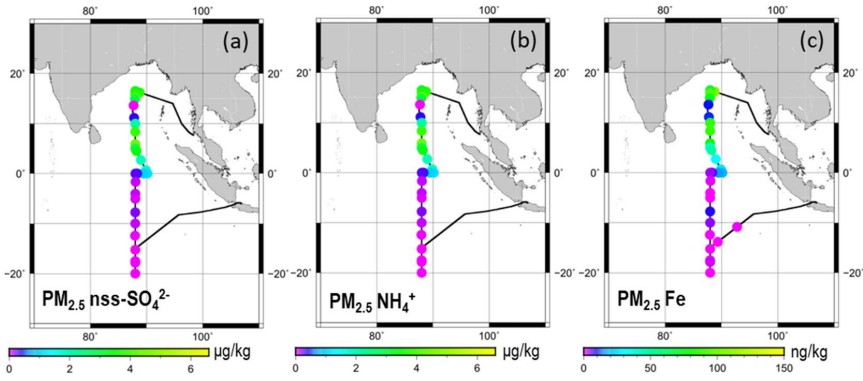

**Figure 2: Horizontal variation of mass concentrations of (a) nss-SO$_4^{2-}$, (b) NH$_4^+$, and (c) Fe in PM$_{2.5}$. Each sample was collected continuously 12 h or 24 h under controlling by wind sector. Data are as shown at averaged locations for latitude and longitude during the sampling period.**



**Table 2: Average, 25th percentile and 75th percentile values of PM$_{2.5}$ ion concentration (μg/m$^3$) at 87–90°E**

|  | 0–16°N | | | 0–20°S | | |
|---|---|---|---|---|---|---|
|  | average | 25th percentile | 75th percentile | average | 25th percentile | 75th percentile |
| Cl$^-$ | 0.05 | 0.02 | 0.04 | 0.21 | 0.03 | 0.30 |
| NO$_3^-$ | 0.07 | 0.04 | 0.09 | 0.06 | 0.04 | 0.08 |
| SO$_4^{2-}$ | 13.07 | 8.32 | 19.10 | 1.50 | 0.48 | 1.67 |
| Na$^+$ | 0.23 | 0.19 | 0.27 | 0.21 | 0.08 | 0.35 |
| NH$_4^+$ | 3.74 | 2.24 | 4.87 | 0.31 | 0.03 | 0.35 |
| K$^+$ | 0.53 | 0.19 | 0.78 | 0.05 | 0.01 | 0.07 |
| Mg$^{2+}$ | 0.04 | 0.03 | 0.05 | 0.03 | 0.02 | 0.04 |
| Ca$^{2+}$ | 0.07 | 0.04 | 0.09 | 0.04 | 0.03 | 0.06 |
| nssSO$_4^{2-}$ | 13.01 | 8.25 | 19.03 | 1.45 | 0.40 | 1.63 |
| nssK$^+$ | 0.52 | 0.19 | 0.77 | 0.04 | 0.00 | 0.06 |
| nssCa$^{2+}$ | 0.06 | 0.04 | 0.08 | 0.03 | 0.02 | 0.05 |

**Table 3: Average, 25th percentile and 75th percentile values of PM$_{2.5}$ metal concentration (ng/m$^3$) at 87–90°E**

|  | 0–16°N | | | 0–20°S | | |
|---|---|---|---|---|---|---|
|  | average | 25th percentile | 75th percentile | average | 25th percentile | 75th percentile |
| Na | 363.0 | 274.0 | 446.0 | 265.4 | 86.3 | 416.8 |
| Al | 141.0 | 57.0 | 164.0 | 15.7 | 0.0 | 31.8 |
| K | 526.8 | 194.0 | 550.0 | 46.1 | 8.6 | 69.7 |
| Ca | 44.1 | 20.6 | 50.2 | 9.4 | 5.5 | 14.7 |
| Ti | 5.9 | 2.3 | 7.3 | 0.3 | 0.0 | 0.6 |
| V | 1.8 | 0.8 | 2.3 | 0.5 | 0.0 | 0.8 |
| Fe | 78.9 | 31.4 | 101.0 | 5.1 | 0.1 | 8.0 |
| Ni | 0.8 | 0.4 | 1.0 | 0.1 | 0.0 | 0.2 |
| Zn | 78.9 | 32.6 | 84.5 | 3.8 | 0.0 | 4.2 |





Among metals measured using ICP-MS, the Na and K mass concentrations were high (71–1280 ng m$^{-3}$ and 6–1210 ng m$^{-3}$, respectively). The Fe concentrations were high (31–162 ng m$^{-3}$) north of the equator, but low (<22 ng m$^{-3}$) south of the equator. The Fe mass concentrations were well correlated positively with nss-K$^+$ ($R^2$ =0.95), nss-SO$_4^{2-}$ ($R^2$=0.93), and Ca ($R^2$=0.90) mass concentrations (Fig. S2). The concentrations of V and Ni, which originate from heavy oil combustion by ships, were correlated well ($R^2$ of 0.96). However, their correlations

with Fe concentration ($R^2$ values of 0.69 and 0.83, respectively) were weaker than the correlation between V and Ni and the correlations of Fe found with nss-K$^+$, nss-SO$_4^{2-}$, and Ca. These results suggest that a large fraction of the observed Fe had been transported from around the continental atmosphere with dust, nss-K$^+$, and nss-SO$_4^{2-}$. However, the good correlation with continental elements implies that Fe was transported together with the continental air mass, but does not imply that emission sources for each element are the same. Therefore,

details of the composition and morphological features of individual Fe-containing particles were investigated as described in the following sections.

## 3.2 Individual particle features and co-existing states with Fe of sulfate and soot

Electron microphotographs of samples #01 and #07 are presented in Fig. 3 as examples. Photographs of all samples (#01–07) and number fractions of the morphological types observed in our samples are also shown

respectively as Figs. S3 and S4 of Supplemental Materials. The morphological types were classified based on a report by Ueda et al. (2016) and features of particles on the samples. For samples #01–06, many particles in all samples had a rounded shape (ball) or were clustered into ball shapes (cluster), such as the blue and white arrowed particles of sample #01 of Fig. 3. Dome-shaped particles have less height to area (dome). Chain aggregation of small globules (ca. 30 nm) is regarded as a particle composed mainly of soot (soot), although

they were a small fraction. For sample #07, some particles constructed of cubic parts having high contrast to an electron beam were also found. They can usually be regarded as sea salt (SS)-shaped.



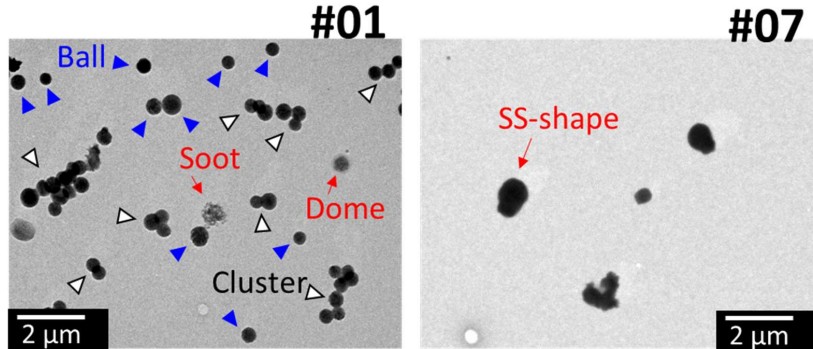

**Figure 3: Electron micrographs for samples #01 and #07. Blue arrowed particles are ball-like shaped particles. White arrowed particles are clusters of balls. Red arrows indicate soot, dome-shaped, and sea-salt (SS)-shaped particles.**

Figure 4a shows a STEM image and EDS mapping of C, O, Al, Si, S, K, Ca, and Fe of an area of sample #03 and examples of the X-ray count spectrum for each particle. Ball-like and clustered particles were composed mainly of S, C, and O, indicating sulfate with organics. Also, K signals in the particles were often found at the same position as the detected S signal. Number fractions of particles detecting each element are depicted in Fig. 5. Also, S was detected from >92% of all particles for samples collected north of the equator. K was detected

from 16–44% particles. For other elements, particles containing sea salt (Na, Mg and Cl) were often detected (3–48% of all particles). Also, Al and Si were detected in about 5% of particles collected north of the equator. Findings show that Fe was detected in 1–4% of all particles.



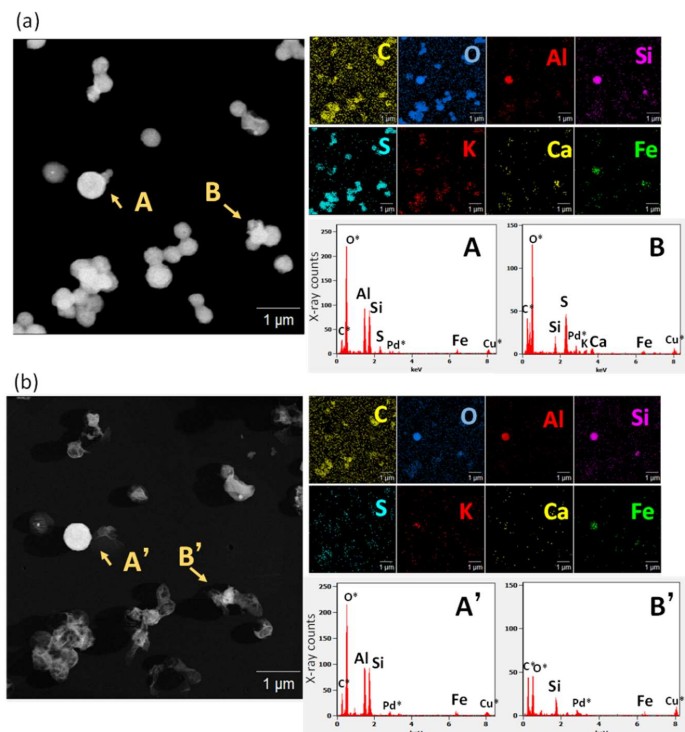

**Figure 4: STEM image and an elemental map of sample #03 for the same region (a) before and (b) after water dialysis and extracted X-ray count spectrum for arrowed particles A and B and for the same region after water dialysis (A' and B'). Asterisked (*) elements represent elements included in the background area without particles.**


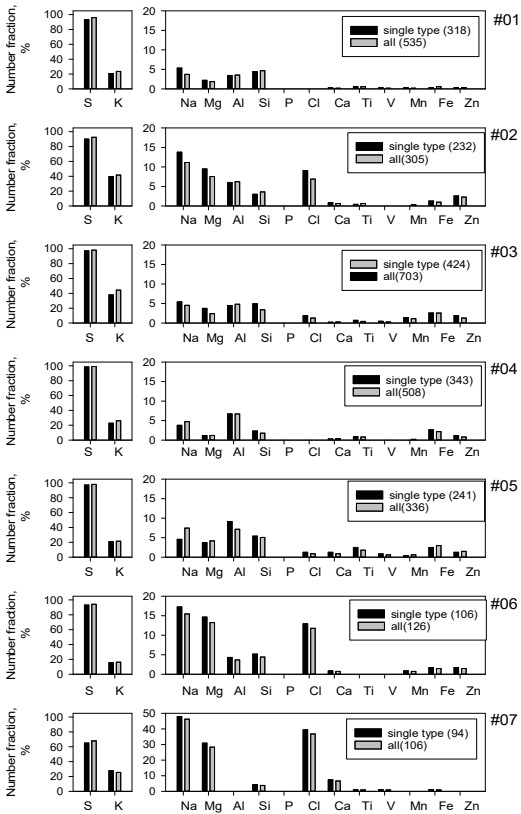


**Figure 5: Number fractions of element-containing particles. Single type is particles except cluster shaped particles. Bracketed numbers in the legend are analyzed particles.**

As shown by particles A and B in Fig. 4a, Fe signals were often found from a partial area with attached or coated sulfate. For this study, metal-congested areas containing Fe are designated as Fe-containing parts, to distinguish

a term from whole particles (Fe-containing particles) such as particles with a co-existing Fe-containing part and sulfate. From some Fe-containing parts, Si, Al and/or Ca were also detected. Compositions of Fe-containing particles and types of Fe-containing parts are detailed in the next section. Figure 4b shows a STEM image, EDS



mapping, and X-ray count spectra after water dialysis for the same area as that shown in Fig. 4a. A large part composed of S and K disappeared with dissolution by water dialysis, suggesting that they have hygroscopicity.

However, many parts composed of Si, Al, Fe or C remained on film.

Co-existing Fe-containing parts and sulfate were found from TEM samples. Such particles are probably formed with secondary formation or coagulation of sulfate with an Fe-containing particle. Good correlation between Fe and nss-$SO_4^{2-}$, as explained in Sect. 3.1, also suggests that a large mass of Fe was transported to the ocean with sulfate particles and their precursor gases. Relations of sulfate particle shapes on samples to their acidity have

been reported from some earlier microscopic studies as explained hereinafter. Acidic sulfate particles such as sulfuric acid ($H_2SO_4$) and bisulfate ($NH_4HSO_4$) are usually found as droplets having a satellite structure because of their property of retaining water even if they were collected after diffusion drying (Waller et al., 1963; Frank and Lodge, 1967; Gras and Ayers, 1979; Bigg, 1980; Ferek et al., 1983; Ueda et al., 2011b). By contrast, particles composed mainly of ammonium sulfate collected after passing the diffusion dryer have been found as

ball-like, rectangular, regular shapes or clusters (Ueda et al., 2011b; Ueda, 2021). Most particles on samples examined for this study, have a ball-like shape or a cluster. This result indicates that sulfates in the particles were more closely related to neutralized sulfate such as ammonium sulfate than to sulfuric acid, which also corresponds with the result found for $PM_{2.5}$, as shown in Sect. 3.1. Particles composed mainly of ammonium sulfate can be present as solid or liquid under their deliquescence humidity, according to atmospheric humidity

experience. In addition, solid ammonium sulfate particles tend to change to rectangular regular shapes when experiencing a metastable humidity condition (Ueda, 2021). However, rectangular particles were rarely observed in this study, although the atmospheric relative humidity (60–81%RH) at the sampling time (Table 1) is usually a metastable condition of ammonium sulfate (35–80%RH). Therefore, sulfate particles in our samples presumably existed as droplet particles in the atmosphere.

Based on reports of measurements of single soot mass analysis by Corbin et al. (2018), much soot originating from combustion of heavy fuel oil in ship engines can contain Fe addition to V, Na, and Pb. Some soot particles



that are less-coated by sulfate were also found from our samples. However, metals including Fe were not detected from less-coated soot particles in this study, as shown in Fig. S5. As reasons for the presence of less-coated soot over the remote ocean, it might be true that hydrophobic or less-coated soot preferentially survives

in the atmosphere after transport (Ueda et al., 2018; Kompalli et al., 2021; Ueda et al., 2022) in addition to local ship's exhaust. For our $PM_{2.5}$ result, as explained in Sect. 3.1, Fe was somewhat correlated with V, although the correlation was weaker than that between V and Ni, or those between Fe and either nss-$SO_4^{2-}$, nss-$K^+$ or Ca. Some Fe might have originated in the ship's exhaust, combined with that from transport from continental areas. However, no strong evidence of the ship's exhaust was found from our individual particle observations,

including analyses of Fe-containing particles presented hereinafter.

### 3.3 Fe-containing particles: composition and morphological features implying their source

Figures 6a and 6b portray number fractions of particles containing the respective elements and mixing types with sulfate for Fe-containing particles (42 particles) observed in five samples (#01–05) affected by air masses

from South Asia. Most Fe-containing particles were mixed with sulfate (Fig. 6b). Actually, S was detected from 90% of Fe-containing particles (Fig. 6a). Also, K was found from half of the Fe-containing particles (Fig. 6a), suggesting that biomass burning affected the Fe-containing particle compositions. However, many K signals were detected from the sulfate coating. Therefore, such K implies secondary formation on Fe-containing parts rather than a source of Fe-containing parts. Both Al and Si were detected from 20% and 30% of Fe-containing

particles. Very little V originating from heavy oil combustion of ships was detected from Fe-containing particles. Moreover, Na, Mg, Ca, Mn, and Zn were detected from less than 7–10% of Fe-containing particles. Both Ca and Mg can originate from biomass burning in addition to sea salt and mineral dust. Adachi et al. (2022) reported ash-bearing particles found in biomass burning smoke based on TEM analysis of fine (<2.5 μm) particles. They defined particles containing both Ca and Mg (>5 weight%) as ash-bearing particles. According to them, ash-



bearing particles from biomass burning smoke commonly have aggregated shapes with complicated
compositions, predominantly calcium with other elements (e.g., C, O, Mg, Al, Si, P, S, and Fe). However, for
our study, the Fe-containing particles mixed with both Ca and Mg were only one (2% of Fe-containing particles).
In addition, morphological features of Fe-containing particles were less like ash-bearing particles reported by
Adachi et al. (2022).

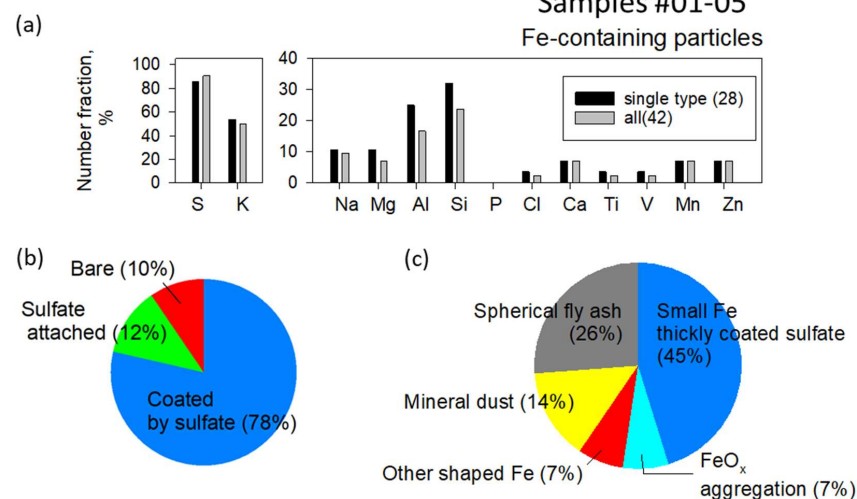


**Figure 6: Mixing states and morphological features of Fe-containing particles of samples collected north of the equator. (a) Number fraction of each-element-containing particles in Fe-containing particles. (b) Pie chart of mixing type with sulfate. (c) Pie chart of Fe-containing part types based on elements and morphology. Section 3.3 presents classification methods of Fe-containing part types. Single type in (a) is particles extracted clustered shaped particles.**

As shown in Figs. 4a and 4b, Fe-containing parts had some different morphological types, such as spherical or
irregular shapes and homogeneous or heterogeneous mixing with other elements. Such morphological types can
be related to their Fe emission source. Based on contrast of STEM images and EDS mapping, we divided Fe-
containing parts to three morphological types associated with an emission source (spherical fly ash, mineral



dust, $FeO_x$ aggregation) and the other two types (other-shaped Fe and small Fe thickly coated by sulfate).

Average ± standard deviation values of *Fe/Pd* for each type above were, respectively, 1.1±2.2, 0.18±0.14, 0.73±0.81, 1.7±0.6, and 0.07±0.05. The number fractions for the morphological types for Fe-containing particles are presented in Fig. 6c. For sample #03, the morphology after water dialysis was also referenced. In addition, examples of STEM images, the elemental map, and X-ray spectra for typical Fe-containing particles are portrayed in Fig. 7. Details for the respective morphological types are explained later in Sects. 3.3.1–3.3.4.

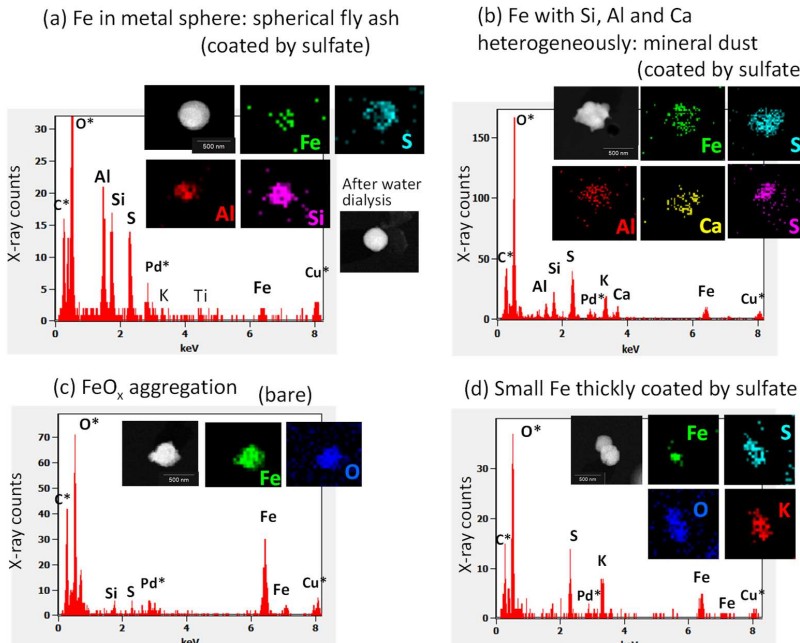


**Figure 7: Examples of STEM images, elemental maps, and X-ray count spectra of typical Fe-containing particles. (a) Example of particle having a Fe-containing part in metal sphere (spherical fly ash). The metal sphere part of this particle is coated by sulfate. (b) Example of Fe-containing particle co-existing with Si, Al or Ca heterogeneously (mineral dust). The Fe, Si, Al, and Ca part of this particle is coated by sulfate. (c) Example of $FeO_x$ aggregation particles. (d) Example of particle having small Fe-**
**containing part thickly coated by sulfate. Asterisked (\*) elements in the X-ray count spectrum represent elements contained in the background area without particles.**



### 3.3.1 Fe in metal sphere (spherical fly ash)

As shown for particle A in Fig. 4 and for the particle in Fig. 7a, Fe-containing spheres with Al and/or Si were found in nine particles (21% of Fe-containing particles) from samples #03 and #04, affected by air masses from the eastern coast of India (Table S1 and Fig. 1b). In addition, Fe spheres without Al or Si were found as two particles (5% of Fe-containing particles) from sample #05, affected by air masses from the Maldives. All particles containing metal spheres of both types co-existed with sulfate. In the STEM image, such metal spheres have high contrast compared to sulfate. Such spheres remained as a residue after water dialysis. Spherical shapes of metals indicate that they were formed through evaporation at high temperatures and subsequent rapid condensation. Moreover, the shape coincides with often encountered features of fly ash particles originating from coal combustion at power plants (e.g. Fisher et al., 1978; Yao et al., 2015; Umo et al., 2019). For this study, metal spheres composed mainly of Fe, Si or Al in our atmospheric aerosol samples are designated as spherical fly ash to distinguish them from fly ash related to emission sources that are not classified using morphology. In coal, Fe generally exists in two forms: as pyrite or aluminosilicate (Tomeczek and Palugniok, 2002). Aluminosilicate-Fe such as kaolinite and illite generally undergoes high-temperature melting and fragmentation, leading to formation as fly ash (Rathod et al., 2020). Although low-temperature combustion processes such as residential uses might not engender Fe volatilization (Flagan and Seinfeld, 2012), power plants can emit fly ash efficiently (Rathod et al., 2020). Although the mass concentration peak of fly ash emitted from power plants is super-micrometric (1–10 μm), submicrometer fly ash particles are also emitted together. They have been observed and described in the literature (Markowski and Filby, 1985; Liu et al., 2018; Umo et al., 2019).

### 3.3.2 Fe co-existing with Al, Si or Ca heterogeneously (mineral dust)

As shown for particle B in Fig. 4 and for the particle in Fig. 7b, some non-spherical parts containing Fe, Si, and Al were also found. Unlike the Fe-containing sphere parts explained earlier, Ca was usually detected from such non-spherical Fe-containing parts. Additionally, they have some domains of different concentrations of Fe, Al,



Si, and Ca. The Fe, Al, Si and Ca are main elements of silicate minerals. Mineral dust is usually composed of some different mineral species domains (Conny, 2013; Jeong and Nousiainen, 2014; Jeong et al., 2014 and 2016; Conny et al., 2019; Ueda et al. 2020). Therefore, heterogeneous structures such as Fe, Al, Si and Ca imply that they are mineral dusts without any origin related to combustion. Such Fe-containing parts regarded as mineral dust were found in six particles (14% of Fe-containing particles) in samples #01, 03, 04, and 05 (Table

S1). Most particles of this type (five particles) co-existed with sulfate. Based on water dialysis of sample #03, Ca in particle B in Fig. 4a dissolved with water dialysis, as shown in Fig. 4b. Although calcite in dust is poorly soluble, chemical transformation from them into calcium sulfate or calcium nitrate with atmospheric aging can alter their solubility (Okada et al., 1990, 2005; Zhang and Iwasaka, 1999; Matsuki et al. 2005).

### 3.3.3 $FeO_x$ aggregation

Some aggregate components comprising Fe and O without other metals were found as shown for a particle in Fig. 7c. They are regarded as being $FeO_x$, such as magnetite and illite. Similar aggregated $FeO_x$ nanoparticles have often been reported from several observational studies of urban atmospheres using electron microscopy methods (Hu et al., 2015; Adachi et al., 2016; Ohata et al., 2018), roadside environments (Sanderson et al., 2016), and polluted remote seas (Li et al., 2017). Aggregated $FeO_x$ co-existing with soot was also found at urban

sites (Ohata et al., 2018; Ueda et al., 2022). From this study, $FeO_x$ aggregation (classified as $FeO_x$ aggregation) was found as without a C-rich part from three particles (7% of Fe-containing particles) in samples #02, 03, and 04, affected by air masses from India (Table S1 and Fig. 1). Two $FeO_x$ aggregations co-existing with sulfate were found. As explained in Sect. 3.2, soot-containing metals were also minor. The $FeO_x$ can be emitted from blast furnaces at Fe-working facilities (Machemer, 2004) and as exhaust from motor vehicles (Kukutschová et

al., 2011; Liati et al., 2015). However, considering $FeO_x$ without soot, $FeO_x$ aggregations examined in this study might have originated mainly from the former source.



### 3.3.4 Other-shaped Fe and small Fe thickly coated by sulfate

For Fe-containing parts of three particles (7% of Fe-containing particles), different shapes were found in terms of the features above: spherical fly ash, mineral dust, and $FeO_x$ aggregation. They were classified as other-

shaped Fe. Of them, two particles were without sulfate. For the other 19 Fe-containing particles, discriminating the shapes of Fe-containing parts was difficult because such parts were too small compared to the sulfate coating, as illustrated in Fig. 7d. Such Fe-containing particles were classified as small Fe thickly coated by sulfate separately from other shaped Fe. The number fraction was 45% of the Fe-containing particles. However, the *Fe/Pd* values (0.07±0.05) from such small Fe were usually less than those of the other types.

**3.4 Fe simulation for each source by the global model**

To elucidate the origins of Fe and the model performance, Fe simulation results using the CAM5-ATRAS model were compared with observation results. Figure 8 shows the monthly mean simulated mass concentrations in $PM_{2.5}$ for total Fe and Fe from each source at the surface. The color scale of 0–150 ng/kg is the same as that presented for Fig. 2c. Averages of the total Fe mass concentrations for areas north and south of the equator

along the ship tracks (0–16°N and 0–20°S at 87.5°E) are shown in Fig. 9a with the observed values. Although the simulated Fe mass tends to be lower than the observed Fe mass, the model simulations well reproduce the contrast of Fe mass between the north (high concentration) and south (low concentration) during the cruise.



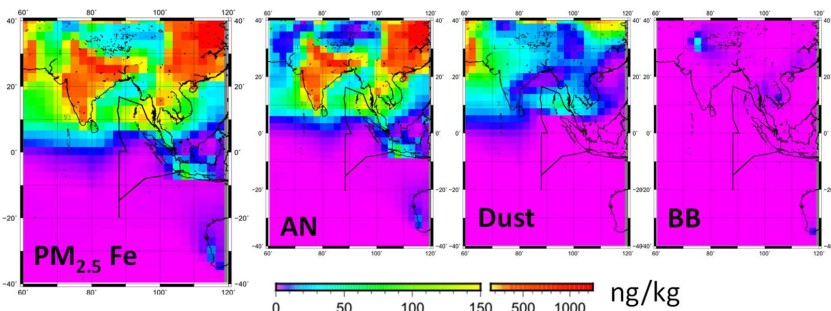

**Figure 8: Spatial distribution of total, anthropogenic (AN), mineral dust (Dust), and biomass burning (BB) iron concentration of PM$_{2.5}$ simulated by CAM5-chem/ATRAS2 model. Values are monthly averaged mass concentrations for November 2018.**

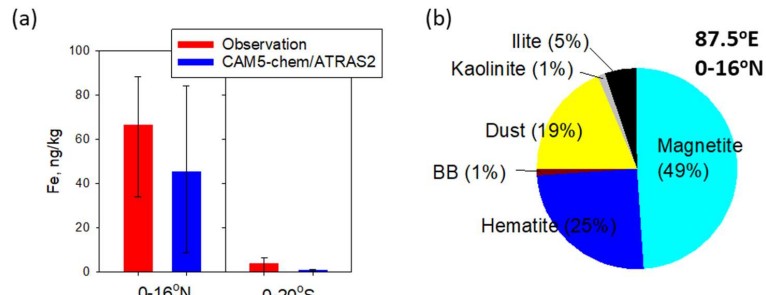

**Figure 9: PM$_{2.5}$ Fe mass concentration simulated by CAM5-chem/ATRAS2. (a) Averaged PM$_{2.5}$ Fe simulated value for 87.5°E and observed PM$_{2.5}$ Fe of 87–90°E during the KH18-06 cruise. The lower and upper error bars respectively stand for the 25th and 75th percentiles. (b) Pie chart of averaged mass fraction simulated PM$_{2.5}$ Fe for each Fe species of 0–15°N at 87.5°E. Simulated values are monthly averaged mass concentrations for November 2018.**

The CAM5-ATRAS results show that the anthropogenic Fe was dominant in PM$_{2.5}$ Fe concentration within the area shown in Fig. 8. The averaged mass fraction of each Fe source/mineral type in the model for 0–15°N at 87.5°E estimated are depicted in Fig. 9b. The anthropogenic Fe (magnetite, hematite, illite, and kaolinite), dust Fe and biomass-burning Fe were, respectively, 80%, 19%, and 1% of the mass of PM$_{2.5}$ Fe. Especially, anthropogenic Fe around India has a high concentration in Fig. 8. The anthropogenic activity of the area might have affected Fe concentrations over the Indian Ocean.



The anthropogenic Fe was mostly (74% in mass of $PM_{2.5}$ Fe) estimated as $FeO_x$ (magnetite and hematite). The mass fraction of anthropogenic aluminosilicates Fe (illite and kaolinite), which derive mainly from coal

combustion, was only 6% in the model. These simulation results might underestimate the fraction of aluminosilicate Fe and overestimate the fraction of $FeO_x$ compared to the TEM results. In the TEM samples, spherical fly ash having sufficient *Fe/Pd* values were often found (26% of Fe-containing particles) as shown in Fig. 6. Also, most of them (8 in 11 spherical fly ash particles) contained Al and/or Si. Mineralogy of Fe from coal combustion in emission inventory by Rathod et al. (2020), which is used in our model, was estimated from

reference to mineralogical measurements of bulk samples using Mossbauer spectrometry or the X-ray absorption near-edge structure. Although the mineralogical transformation from fuels to by-product can be affected by fuels and combustion temperatures, the Fe contents estimated from mineralogical measurements of coal fly ash at power plants tend to comprise both oxides (such as magnetite and hematite) and clay (such as kaolinite and illite) (Hinckley et al., 1980; Szumiata et al., 2015; Waanders et al., 2003; Oakes et al., 2012;

Rathod et al., 2020). However, even if Fe in spherical fly ash in our TEM samples were assumed to be composed of aluminosilicates and equivalent $FeO_x$, the mass fraction of aluminosilicates Fe inferred from TEM results would be larger than 13%, which is higher than our model simulated results. The chemical composition and size-mass distribution of coal fly ash particles at the source can be affected by multiple factors such as the fossil fuel composition and combustor (Markowski and Filby, 1985; Liu et al., 2018). Their lack of clarity can also

affect the uncertainty of simulation for the emission of Fe originated in coal combustion and the transport according to the particle size.

**3.5 Water solubility of Fe**

Figure 10 portrays the relation between *Fe/Pd* before and after water dialysis and $f_{WSFe}$ for Fe-containing particles (14 particles) for sample #03. The symbols were made according to the composition and shape of

residues after water dialysis. For particles containing metal sphere residues regarded as fly ash, the values of





Fe/Pd after water dialysis of Figs. 10a and 10b correspond well to the values before water dialysis at a 1:1 ratio. Because Pd mass after water dialysis can be regarded as equal to that before water dialysis, such a 1:1 relation indicates that the Fe in spherical fly ash was almost insoluble in water. For particles aside from metal spheres containing residues, the detected values of *Fe/Pd* decreased, suggesting that some Fe were soluble in water. The

averages of $f_{WSFe}$ were 48%, 6%, 58%, and 65%, respectively, for all Fe-containing particles of spherical fly ash, mineral dust, and Fe-containing particles, except spherical fly ash (Fig. 10c).

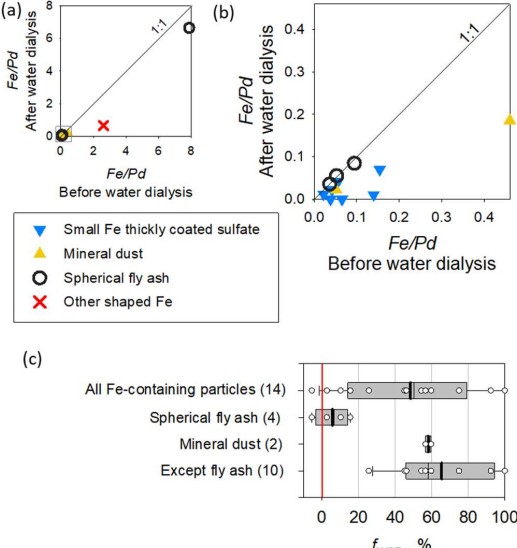

**Figure 10: Relations of water-soluble and water-insoluble Fe in individual Fe-containing particles on sample #03. (a) Scatter plot of Fe weight normalized Pd before and after water dialysis. Pd is a coating material in alloy for shadowing to samples. (b)**
**Enlarged graph of the light-gray squared area of (a). (c) Box plot of water-soluble Fe fraction ($f_{WSFe}$). The lower boundary of the box shows the 25th percentile. The line within the box represents the median. The upper boundary of the box stands for the 75th percentile. Whiskers above and below the box respectively show the 90th and 10th percentiles. White circles show values for the respective particles.**



Some measurement studies for bulk aerosol samples have reported fractions of water-soluble Fe in all Fe
(Kumar and Sarin, 2010; Baker et al., 2013; Ingall et al., 2018). After Kumar and Sarin (2010) also measured
Fe in $PM_{2.5}$ at a high-altitude site in a semi-arid region of western India, they reported fractions of water-soluble
Fe as 0.06–16.1%. Ingall et al. (2018) measured the water solubility of total Fe in bulk aerosol samples taken
from multiple locations in the Southern and Atlantic oceans, Noida (India), Bermuda, and the eastern
Mediterranean (Crete). Their fractions of water-soluble Fe were low (<5%) under samples influenced by
Saharan dust, but the fractions for samples of Noida (3–20%) and samples influenced from Europe were high
(17–35%), indicating an anthropogenic contribution of soluble Fe. From measurements taken of the remote
ocean, Baker et al. (2013) reported Fe solubility 2.4–9.1% of aerosol collected in the Atlantic Ocean during
research cruises. Compared to Fe solubility by the studies described above, the water-soluble Fe fraction for
individual Fe-containing particles in this study tended to be estimated as higher. This finding might be attributed
to the fact that our analyzed samples were smaller particles (0.3–0.8 μm samples stage diameter), with a greater
surface relative to the particle mass, and aged particles collected over the remote ocean, as explained below.

The values of $f_{WSFe}$ in this study were 20–100% in Fe-containing particles, except spherical fly ash. Among Fe-
containing minerals, the fraction of water-soluble Fe in $FeO_x$ (magnetite, goethite and hematite) and clays (such
as kaolinite and illite) are very low (respectively, <0.01% and 1.5–4%), but that of ferrous and ferric sulfate is
higher (50–90%) (Desboeufs et al., 2005; Schroth et al., 2009; Journet et al., 2008; Rathod et al., 2020). The Fe
solubilities for many Fe-containing particles in this study were comparable to those of ferrous and ferric sulfate.
Our earlier study often found non/less-sulfate-coated $FeO_x$ in aerosol samples collected in urban Tokyo, for
which we applied the same analysis (Ueda et al., 2022). However, few decreases of Fe with water dialysis were
found. The difference of Fe solubility found from this urban study might be attributed to atmospheric aging
processes by coating of secondary aerosol materials and Fe oxidation during transport. Some experimentation
and simulation studies have indicated that Fe in minerals composed of Fe oxide and clays can be oxidized and
enhanced in acidic liquid phase such as aerosol droplets and clouds (Shi et al., 2009 and 2015; Chen et al., 2012).



For the present study, most of the observed Fe-containing particles co-existing with sulfate were regarded as droplets in the atmosphere. Therefore, Fe in particles might have oxidized in liquid droplet particles in the atmosphere, after which they were later collected as water-soluble Fe, such as Fe sulfate. For Fe-containing particles in this study, the number fraction of $FeO_x$ aggregation was less, whereas that of categorized particles to small Fe thickly coated by sulfate was higher. This result might also be affected somewhat by the loss of the water-insoluble Fe shape by change to water-soluble Fe.

All water insoluble residues in Fe-containing particles of $f_{WSFe}<15\%$ were spherical fly ash. Although all of them co-existed with sulfate, water dialysis results indicated that the water insolubility of Fe in them was retained. Our observed insoluble sphere residues imply structural and morphological features as reasons for the insolubility of the Fe contained in aged fly ash particles. Spherical particles have the minimum surface area. In addition, the Fe is distributed in particles with other insoluble materials composed of Si and Al in the sphere. This distribution would physically block the oxidation and dissolution of Fe. However, results of several studies have suggested that Fe in similar spherical fly ash particles can dissolve in acidic particles (Chen et al., 2012; Li et al., 2017). Li et al. (2017) observed similar submicrometer Fe-rich spheres coated by sulfate in samples collected over the yellow sea affected by the East Asian continental outflow. They analyzed sulfate-rich particles containing Fe-rich parts using nanoscale secondary ion mass spectrometry and elemental mapping with STEM, and reported the presence of dispersed $FeS^-$ (Fe sulfate) around $FeO^-$ (Fe oxide)-rich part. They concluded that such Fe sulfate was formed from Fe dissolution of fly ash in an acidic aqueous phase because no other atmospheric source of Fe sulfate or process leads to its formation. Using bulk samples of coal fly ash composed mainly of spherical particles, Chen et al. (2012) also investigated Fe dissolution. Their experiments demonstrated that Fe in coal fly ash can dissolve in $H_2SO_4$ acidic aqueous solutions of pH 1 and 2. These earlier reported study results suggest that some Fe in fly ash can exist as water-soluble Fe in sulfuric acid particles. However, results of our experiments indicated that soluble Fe in spherical fly ash was considerably less or nonexistent compared to insoluble Fe. From the present study, as shown in Sect. 3.2, sulfate particles were

found as particles neutralized, such as by ammonium. Such atmospheric conditions observed in the present study might not have decreased pH sufficiently to enhance the Fe solubility of fly ash.

Our observation results of TEM samples suggest fly ash as an important component of fine Fe-containing
particles. In addition, water dialysis results suggest that Fe in aged particles over the remote ocean tend to exist as partly or mostly water soluble, whereas Fe in spherical fly ash can maintain water-insolubility. Model simulations have led to estimates that $FeO_x$ was a major component in $PM_{2.5}$ Fe of this area and that Fe from anthropogenic aluminosilicates from sources such as coal combustion was minor. Underestimation of the mass fraction of Fe keeping water insoluble during atmospheric transport can be a factor affecting the estimation error
of water-soluble Fe deposition. In addition, although Fe mineralogy is often used for the simulation of solubility of anthropogenic Fe in percent in some models, the spherical shape of fly ash particles and the results of water dialysis imply that Fe solubility and the change in the atmosphere depend not only on the mineralogical but also the morphological and structural features of fly ash originating from emission processes. The presence of fly ash in fine aerosols should be noted for model simulation of water-soluble Fe and estimations based on size-
segregated samples of Fe concentrations.

**4 Summary and Conclusions**

To elucidate the mixing states and water solubility of Fe-containing particles in remote marine areas, we conducted ship-borne aerosol observations over the Indian Ocean during the R/V Hakuho Maru cruise. After
TEM samples for individual particle analysis were obtained, they were also analyzed using EDS and water dialysis. Most of the particles were composed mainly of sulfate neutralized by ammonium or potassium: the particle number fraction, 0.6–3.0%, of particles on a sample stage of 0.3–0.8 µm diameter contained Fe. They mostly co-existed with sulfate.



Backward air mass trajectory analyses suggest that air masses south of the equator were transported from

southern India. Both the correlations of the respective elements measured using chemical analysis for bulk $PM_{2.5}$ samples and the absence of V and soot for individual Fe-containing particles implied that the Fe in particles was transported mainly from around the continent rather than from ship exhaust. The Fe in particles was found to be 26% metal spheres, often co-existing with Al or Si, regarded as fly ash, 14% as irregularly shaped heterogeneously co-existing with Si, Al or Ca, regarded as mineral dust, and 7% as $FeO_x$ aggregations.

Global model simulations using a recent emissions inventory mostly reproduced the observed $PM_{2.5}$ Fe concentrations in the north and south during the cruise. The model simulations suggested that $PM_{2.5}$ Fe over the observation site was influenced strongly by anthropogenic Fe emissions around India. In contrast, compared with the morphological features of observed Fe, the simulations tend to overestimate the fractions of anthropogenic $FeO_x$ and to underestimate the fraction of Fe in aluminosilicates originating from coal

combustion.

Water-dialysis analysis for a TEM sample indicated that about half of the Fe in Fe-containing particles was soluble in water. However, Fe in spherical fly ash particles was almost insoluble in water even when co-existing with sulfate. Dissolution and oxidation of Fe in spherical fly ash might have been blocked by the small surface of the sphere and the structure of Fe-dispersion in other insoluble aluminosilicates.

Our results obtained from shipboard observations and individual fine aerosol analysis indicate that Fe of various types, such as fly ash, $FeO_x$, and mineral dust, coexist with sulfate over the remote Indian Ocean, and indicate that their solubilities differ among types. Although the model simulations show good agreement with the observed Fe mass concentrations, we also find a marked difference in the mass fractions of mineral sources of model simulations compared to observed Fe types. Earlier studies have indicated that anthropogenic fine Fe

tends to be composed mainly of $FeO_x$, with increased solubility occurring along with aging. However, our results suggest that Fe in spherical fly ash can stand out in fine aerosols over the remote ocean and maintain water insolubility. For accurate estimation of the effects of atmospheric Fe on marine biogeochemical activity, more



proper attention must be devoted to morphological and mineral types of Fe depending on the source, especially to insoluble Fe in fly ash.


## Acknowledgments

We are indebted to staff members of the Hakuho Maru for assisting our work on board and Prof. K. Osada of Nagoya University for support and technical advice. We also extend my gratitude for technical support from the High Voltage Electron Microscope Laboratory of Nagoya University. We gratefully acknowledge the

NOAA Air Resources Laboratory (ARL) for providing the HYSPLIT transport model (http://www.arl.noaa.gov/ready.html). This study was supported by the Ministry of Education, Culture, Sports, Science, and Technology and the Japan Society for the Promotion of Science (MEXT/JSPS) KAKENHI Grant Numbers 18H03369, 18J40204, 19K20438, 22K18023, JP19H04253, JP19H05699, JP19KK0265, JP20H00196, JP20H00638, JP21K12230, JP22H03722, and JP22F22092, MEXT Arctic Challenge for

Sustainability phase II (ArCS-II; JPMXD1420318865) projects, and the Environment Research and Technology Development Fund 2–2003 (JPMEERF20202003) of the Environmental Restoration and Conservation Agency. M.L. acknowledges the support of JSPS Postdoctoral Fellowships for Research in Japan (Standard).

## Data availability

Back-trajectory data were calculated from the NOAA HYSPLIT model

(https://www.ready.noaa.gov/HYSPLIT.php, last access: 08 March 2018). Other data will be provided upon request.



**Author contributions**

SU, YI and FT designed the study. SU analyzed TEM samples and wrote the paper. YI worked on board for aerosol sampling and measurement. YI and FT contributed to chemical analyses of $PM_{2.5}$ samples. HM and ML conducted numerical model simulations using CAM5-ATRAS.

**Competing interests**

The authors declare that they have no conflict of interest.

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
