# Peer review of "Morphological features and water solubility of iron in aged fine aerosol particles over the Indian Ocean"

_EGUsphere, 2023_

## Author Comment (AC1)

*Reviewer #1 Comments:*

*This manuscript by Ueda et al., investigated the morphological features, mixing states and water solubility of Fe-containing particles in aged fine aerosol particles over the Indian Ocean. The topic of this study is attractive and interesting, which is very useful for understanding the ageing of particles and the dissolution of Fe-containing particles during the transport of aerosols and the morphology variations of Fe-containing particles. Overall, the manuscript is logical, and the main issues are very well discussed in this paper. I would therefore recommend this manuscript for publication after the authors have addressed the following comments.*

Response:

We appreciate the many constructive comments offered by Prof. Weijun Li, which have improved our manuscript considerably. Revisions have been highlighted as red in the text of the revised manuscript. This manuscript was checked according to the journal guidelines by a native-English speaking professional with experience in the review of technical documents in this field.

*Comments:*

*Major concerns:*

1. *The units of horizontal variation of mass concentrations of (a) nss-SO42-, (b) NH4+, and (c) Fe in PM2.5 in Figure 2 are μg/kg and ng/kg, however, there are μg/m3 and ng/m3 in Table 2, which is confused for me, please explain the differences between the two units and make them uniform.*

Response:

Those mass concentrations were measured in units of $\mu g/m^3$ and $ng/m^3$. For Figure 2, the Fe mass concentrations are shown as a mixing ratio (μg/kg and ng/kg), which facilitates readers' comparison to model output data presented in Figure 8. Mixing ratios were calculated using the daily average of temperature and atmospheric pressure measured onboard. We added the explanation presented above to section 3.1. The units in the revised Tables 2 and 3 were changed to present units of μg/kg and ng/kg uniformly.

*Comments:*

2. *Figure 2c (mass concentrations of Fe) has two more data points compared to figures 2a and 2b. Why?*

Response:

As explained in section 2.2, we measured ions (such as (a) nss-$SO_4^{2-}$ and (b) $NH_4^+$) using samples collected on a quartz fiber filter, and metals (such as Fe (c)) using samples collected on Teflon filter. Because stocks of quartz fiber filters were finished up, the sampling was stopped before that using a Teflon filter. We added some mention of each sampling period to section 2.2 of the revised manuscript.

*Comments:*

3. *Line 232: "For non-sea-salt components, the relations between the doubled nss-SO42− molar concentration and the NH4+ plus nss-K+ molar concentration were usually between 1:1 and 2:1 (Fig. S1b), suggesting that nss-SO42− originated from ammonium sulfate, ammonium bisulfate, and ammonium potassium rather than from sulfuric acid." Here, "ammonium potassium" should be "potassium sulfate" ?*

Response:

Thank you for noticing that mistake. That was corrected to potassium sulfate.

*Comments:*

4. *Line 227-228 and line 232-233: "The values nss-K+, which are regarded as originating mainly from biomass burning……; For non-sea-salt components, the relations between the doubled nss-SO42− molar concentration and the NH4+ plus nss-K+ molar concentration were usually between 1:1 and 2:1". The author argued that nss-K+ was mainly from biomass burning, please provide more evidence to support this view.*

Response:

As already described, nss-$K^+$ has often been used as a tracer of biomass burning (Andreae, 1983; Kawamura and Kaplan, 1987; Narukawa et al., 1999). In addition, highly frequent agricultural burning around the windward area of our observation site in autumn was reported from some studies (Bray et al., 2019; Shaik et al., 2019). Therefore, we thought that the air mass of the north site in this study might have been affected by biomass burning around the windward continental area. However, we thought that the other possibilities should not be denied completely. We carefully revised the description with references as shown in the first paragraph in section 3.1.

*Comments:*

5. *In the section: "3.2 Individual particle features and co-existing states with Fe of sulfate and soot",*

*the author chooses samples #01 and #07 as example to illustrate the individual particle features, however, I fail to get the idea why the author selects these two samples. Is there any special in these two samples, the reasons should be given.*

Response:

Because morphological features of samples #01-06 were similar, we used sample #01 as a representative example, and sample #07 as the other examples. Figures for photographs and morphological types of all samples are presented in supplemental materials (Figs.S3 and S4 before revision). In the revised manuscript, to show similarity of samples #01-06, the graph for morphological types was moved to Figure 3, with some revision of the first sentences in section 3.2.

*Comments:*

6. *Compared with the filed observation results, the CAM5-ATRAS model underestimates Fe by nearly 1/3 (from Figure 9a), why? Are there other unknown sources of Fe or is there a large uncertainty in the Fe emission inventory used by your model?*

Response:

As you have commented, a great amount of uncertainty exists in Fe emission inventories from all sources including anthropogenic materials, dust, and biomass burning origin. While greater underestimation of Fe exists in earlier model estimation, our model has improved Fe simulations with detailed representations of anthropogenic Fe (Matsui et al., 2018b) and new emission data of anthropogenic Fe by Rathod et al. (2020). Although with underestimation of nearly 1/3, our simulations have shown higher Fe concentrations and better agreement with observations than earlier estimates (Liu et al., 2022). The comparison to TEM results suggests that our model underestimates the fraction of anthropogenic aluminosilicate Fe, such as illite and kaolinite. Therefore, their emissions might be underestimated, leading to underestimation of total Fe mass concentrations. Additionally, different spatial and temporal scales between observations and models might also explain the model-observation differences in this study. These explanations were added to section 3.4.

*Comments:*

*Some other minor issues:*

1. *Line 132: $Cl^-$, $NO3^-$, $SO42^-$, $Na+$, $NH+$……., here, "$Cl^-$- and$NH+$" should be revised to "$Cl^-$ and$NH4+$". Please check the similar issues in whole text.*

2. *The numbers and letters are so small that they can't be read clearly in Figure 4 and Figure 7,*

*please adjust the font size.*

Response:

Thank you for pointing out the matters listed above. We rechecked and revised the manuscript.

---

## Author Comment (AC2)

*Reviewer #3 Comment*

*The authors investigate aerosol particles' iron (Fe) properties over the Indian Ocean aboard a research vessel. The article presents important information on these particles' morphology, concentration, and degree of solubility. In general terms, the article is well-written. However, some minor observations need to be addressed. The article can be accepted after answering the following questions.*

Response:

We appreciate the many constructive comments offered by anonymous reviewer #3, which have improved our manuscript considerably. Revisions have been highlighted as red in the text of the revised manuscript. This manuscript was checked according to the journal guidelines by a native-English speaking professional with experience in the review of technical documents in this field.

*Comments:*

*Abstract*

*line 15. the authors mention that they analyzed particles in the size range 0.3-0.8, using a cascade impactor. As written, it appears that they studied a continuous range of particle sizes. However, the methodology mentions that the sampler only has three nominal sizes (1.6, 0.8, and 0.3 µm). Please mention these three sizes in the abstract.*

Response:

TEM samples were collected using a three-stage cascade impactor, but only those of the third stage were used for this study. Particles larger than the 50% cut-off diameter of the stage are collected to efficiency of higher than 50% on the stage. Therefore, in the third stage of the impactor, particles from about the third stage of cut-off diameter (aerodynamic diameter) to about the second stage cut-off diameter were collected. For this study, we specifically examined submicrometer particles that can include many anthropogenic aerosols, and analyzed only the stage samples that mainly submicrometer particles collected. Therefore, we described only the size range of the stage used for analysis as necessary information in the *Abstract*. Although use of only a third stage was also explained before revision, it might have been difficult to understand that point in the *Abstract*. Therefore, we revised the text in the *Abstract*.

*Comments:*

*Methodology*

*The water dialysis process was important to study the water's mixing state and elucidate the Fe's solubility. Could you expand the description of this technique and mention the instrument used for it? Indicate the bibliographic source from which equations 1 and 2 were obtained.*

Response:

Water dialysis method with image analysis had been used for quantification of the volume of water-soluble materials by Okada (1983). This method needs no other special instruments of already written EDS and TEM, although a petri dish and simple dropper are used. For the method described by Okada (1983), they estimated the particle volume before and after water dialysis using image analyses. For our present studies, we tried and applied water dialysis to quantify changes of Fe before and after water dialysis using EDS analysis. Therefore, the experimental method can be cited from earlier studies, but the quantitative method of Fe by the equations is our original. We also used the same method in Ueda et al. (2022), but did not mention the equations. For this study, demonstration of these equations was supported in supplemental materials before revision. In the revised version, we moved the demonstration in the main text (sections 2.3.1 and 2.3.2) and reinforced the description of this technique.

*Comments:*
*Please revise the wording in the first line of the first paragraph of section 2.1. It is not easy to understand the name of the cruise ship where the study was conducted.*

Response:
The sentence was revised as presented below.
Before revision>> Atmospheric observations were conducted over the Indian Ocean during the R/V Hakuho Maru during KH-18-6, which took place on 6–28 November 2018.
After revision>> Atmospheric observations were conducted over the Indian Ocean during the R/V *Hakuho Maru KH-18-6* cruise, which took place on 6–28 November 2018.

*Line 200: Check the spelling in this sentence.*

Response:
The spelling is no problem because the spelling represents the origin of the model's name. We have used the same form in reports of earlier studies (e.g. Matsui, 2018; Liu et al., 2022).

*It is recommended to use the same concentration units in tables and figures. Tables 2 and 3 report the concentrations in μm/m3 and ng/m3, respectively. While in Figure 2, the units are reported as μg/kg.*

Response:

Mass concentrations of ions and metals were measured in units of $\mu g/m^3$ and $ng/m^3$. For Figure 2, the Fe mass concentration is shown as a mixing ratio (μg/kg and ng/kg) for readers to compare them to model output data of Figure 8. Mixing ratios were calculated using the daily average of temperature and atmospheric pressure measured onboard. We added the explanation given above to section 3.1. The units in the revised Tables 2 and 3 were presented uniformly as μg/kg and ng/kg.

*Comments:*
*Results*
*It is suggested that labels (a), (b), (c), and (d) have to be added to each panel in Figure 8. The figure caption should also be improved to make it easier to read.*

Response:

We added the related labels and revised the captions.

*Comments:*
*Figure 10 has many elements. It is suggested to separate them and present some results individually. In addition, the figure caption is very long and difficult to read.*

Response:

We separated the figure to Figures 10 and 11. The caption was also revised.

---

## Author Response (AR2)

*Editor Comments:*

*I thank the authors for addressing most of the comments listed by both reviewers; however, the manuscript requires additional work before it can be accepted for publication in ACP.*

Response:

We appreciate your many valuable comments and criticisms. We have undertaken efforts to improve our manuscript as we can.

*Major/minor comments:*

*1. The authors argue that "This manuscript was checked according to the journal guidelines by a native-English speaking professional with experience in the review of technical documents in this field"; however, I found many grammatical errors along the document. I invite the authors to significantly improve the English of the entire manuscript.*

Response:

Thank you for the comments. We checked the entire manuscript and will continue our efforts. However, it is unfortunately impossible for a non-native speaker of the language to find all errors. Because we understand that well, we have sought the help of reliable native-English speaking professionals after doing our best. The proofreading has been done by multiple native English speakers who have experience at proofreading many papers that have been accepted to international academic journals, including *ACP*, before and since *ACP* was founded in 2001. The present manuscript was also rechecked after sharing this editor comment. Specifically, all of the language requirements for this journal were reviewed and checked. American English spelling and grammar were used, which might be objectionable to speakers from other countries, but which is specifically allowed in the journal guidelines. It would help if you could point to exact grammatical points that you regard as errors. Perhaps specific points can be discussed and resolved. We are committed to improving the manuscript to your high standard. More than anything, we believe that this journal will be fair to the science.

*2. Along the document there are too many qualitative words such as "Presumably", "it might be", and "might have", among others. I invite the authors to avoid such statements and to be more quantitative.*

Response:

We agree that such statements might detract from the quantitative strengths of our presentation. In fact, we have used such words to distinguish supposition and speculation from mathematically probable and certain findings. Therefore, we checked the use of such words (in L334, L339, L343, L467, L529, L534, L593 before revision) critically throughout our manuscript. Although some of the expressions remain because of their context, we tried in most cases to replace or clarify those statements by revision.

*3. I invite the authors to reduce the large number of self citations.*

Response:

We deleted some self-citations and related sentences from the text with this revision (L82, L156–159, L189 and L338–341 before revision).

*4. L247-248: I think this is rather speculative and it is not supported by the present results.*

Response:

We had regarded potassium as an important tracer over the remote ocean based on earlier studies to which we referred before revision. However, we have reconsidered after reading reviewer and editor comments, although they mentioned no clear reason for denying that point. This interpretation seems weak in current studies. Therefore, sentences related to the implication of a relation between potassium and biomass burning were deleted with this revision. This revision does not have important effects on our main conclusions related to Fe-containing particles.

*5. Make units consistent between Fig 2 and Table 2 and 3.*

Response:

The units μg/g for Figures 1a and b, and Table 2 were changed to ng/ng.

*6. Figure 9. Add log scale in panel (a).*

Response:

Because Figure 9a specifically emphasizes the difference between the Northern and Southern

hemispheres, we do not think it is necessary to use a logarithmic axis. The logarithmic axis obscures the difference between the model and the observations for the Northern Hemisphere. Furthermore, we do not think that strict comparison of the difference has strong meaning for the quite low concentrations prevailing in the Southern Hemisphere. Therefore, we did not change Figure 9a. Instead, the average values of Fe concentrations were added to the text (L686).

*Technical comments:*
*L24-25: "aluminosilicate Fe contained in matter such". Please re-write it.*

Response:
The sentence was changed to "anthropogenic aluminosilicate (illite and kaolinite) Fe contained in matter such." in the revised manuscript.

*L26: Do the authors mean "Fe-containg particles"?*

Response:
In this case, we mean the Fe-containing part rather than Fe-containing particles including sulfate coating. Sentences of L26 were revised as presented below.
Before revision>> "Our observations suggest that Fe in particles over remote ocean areas has multiple shapes and minerals, and further suggest that its solubility after aging processes differs depending on their morphological and mineral type."
After revision>> "Our observations revealed multiple shapes and compositions of Fe minerals in particles over remote ocean areas, and further suggested that their solubilities after aging processes differ depending on their morphological and mineral types."

*L37: Add a reference after "areas"*
*L40: Add a reference after "areas"*

Response:
References were added.

*L41: What do the authors mean with "evaporation and condensation processes"*

Response:

The sentence was revised as presented below.

Before revision>> "By contrast, combustion Fe is emitted as both fine and coarse particles, through evaporation and condensation processes"

After revision>> "By contrast, combustion Fe is emitted as both fine and coarse particles through evaporation of metals at higher temperatures in thermal sources and through condensation processes that occur with diffusion and cooling"

*L58: Add a reference after "sources"*

Response:

References were added.

*L59: Add a reference after "Fe"*
*L60: This was mentioned 2 lines above*

Response:

We deleted the sentence of L59 because we thought that the sentence had no strong meaning for this study. Regarding L60, the sentence two lines above mentions Fe at the emission sources. However, the sentence of L60 mentions Fe change with atmospheric aging. Therefore, we did not change the sentence. However, we think that deletion of the sentence of L59 between the other two sentences made it easier to understand the difference (L63–L66 after revision).

*L63: Add a reference after "simulations"*

Response:

This sentence is related to the sentence immediately prior and references. Instead of presenting an added cited reference, the sentence was revised as shown below.

Before revision>> "…; Sakata et al., 2022). Earlier knowledge about relations between solubility, Fe mineral species, and aging processes has usually been based on bulk sample measurements, laboratory experiments, and simulations."

After revision>> "…; Sakata et al., 2022). This earlier knowledge about relations between solubility, Fe mineral species, and aging processes has usually been based on bulk sample measurements,

laboratory experiment findings, and simulation results."

*L65: Add a reference after "insufficient"*

Response:

The sentence was deleted because the meaning overlapped with the last sentences in the section.

*L73: Add a reference after "areas"*

Response:

Cited references were added. The sentence was also changed in minor ways according to the cited reference.

*L113-114: Why is it mentioned " 6–16 November 2018" twice. This is unclear.*

Response:

We are sorry. The second period is 17–19 November 2018. That was corrected in the revised manuscript.

*L128: (a.s.l. 14 m).?*

Response:

We corrected the expression to "located at about 14 m altitude from sea level" in the revised manuscript.

*L132: "the home laboratory"?*

Response:
The words were deleted.

*L143: "Aerosol particles were collected for morphological particle analysis using TEM" What does it mean?*

Response:

We corrected the sentence and the subsequent sentence.

Before revision>> "Aerosol particles were collected for morphological particle analysis using TEM. Aerosols after diffusion drying were collected on carbon-coated nitrocellulose (collodion) films using cascade impactors."

After revision>> "After diffusion drying, aerosols were collected on carbon-coated nitrocellulose (collodion) films using cascade impactors. Then the morphologies and compositions of particles were analyzed using TEM."

*L144: "cascade impactors". Please add model and manufacturer.*

Response:

The cascade impactor used for this study was designed by the authors. Design information such as the materials and nozzle diameters was added to the revised manuscript. The 50% cut-off diameters were calculated using equation 5.28 in Hinds and Zhu (2022, Third edition Aerosol Technology, Wiley, pp109). The relevant reference was also added to the revised text. With this revision, we found correction of the Stokes number from the first edition of Hinds (1982), from 0.22 to 0.24. Therefore, the calculated result of the 50% cut-off diameter of second stage corrected from 0.8 μm to 0.9 μm.

*L146: "flow rate of 1.0 L min−1." It this correct? This looks pretty low to me.*

Response:

It is correct. Such a flow rate is well used in the TEM sampler, which has a single nozzle (e.g. Kojima et al., 2006; Neimi et al., 2006; Li et al., 2013; Adachi et al., 2020). The cutoff diameter of the impactor depends mainly on the flow velocity through a jet nozzle to the impaction plate. As an impactor of the higher flow rate, there is an impactor having multiple orifices, such as the MOUDI series. Such a multiple orifice impactor is designed to collect bulk particles to constitute a detectable amount for chemical analysis rather than for TEM analysis. Such a sampler also seems to be capable of use as a TEM sampler with some arrange holder. However, it does not require the accumulation of numerous particles for individual particle observation because dispersed particles on film are better

for observation. In addition, the TEM grid is small (5 mm diameter). Therefore, we think that a single-nozzle sampler with low flow has more generally been used for aerosol sampling for TEM analysis.

References

Niemi, J. V., Saarikoski, S., Tervahattu, H., Mäkelä, T., Hillamo, R., Vehkamäki, H., Sogacheva, L., and Kulmala, M.: Changes in background aerosol composition in Finland during polluted and clean periods studied by TEM/EDX individual particle analysis, Atmos. Chem. Phys., 6, 5049–5066, https://doi.org/10.5194/acp-6-5049-2006, 2006.

Kojima, T., Buseck, P. R., Iwasaka, Y., Matsuki, A., Trochkine, D.: Sulfate-coated dust particles in the free troposphere over Japan, Atmos. Res., 82, 698-708, https://doi.org/10.1016/j.atmosres.2006.02.024, 2006.

Li, W., Wang, T., Zhou, S., Lee, S., Huang, Y., Gao, Y., Wang, W.: Microscopic observation of metal-containing particles from Chinese continental outflow observed from a non-industrial site, Environ. Sci. Technol., 2013, 47, 16, 9124–9131, https://doi.org/10.1021/es400109q, 2013.

Adachi, K., Dibb, J. E., Scheuer, E., Katich, J. M., Schwarz, J. P., Perring, A. E., Braden Mediavilla5, Hongyu Guo4,6 , Campuzano-Jost, P., Jimenez, J. L., Crawford, J., Soja, A. J., Oshima, N., Kajino, M., Kinase, T., Kleinman, L., Sedlacek, A. J., Yokelson, R. J., Buseck, P. R.: Fine ash-bearing particles as a major aerosol component in biomass burning smoke. J. Geophys. Res.: Atmos., 127, e2021JD035657. https://doi.org/10.1029/2021JD035657, 2022.

*L224: Is "water" an aerosol species?*

Response:

Liquid water in the atmosphere is fundamentally present as water in hygroscopic aerosols or clouds and rain droplets. For example, for chemical analyses of bulk aerosol samples, we also agree that water and the other species should have separate explanations. However, in this case, the sentence explains the aerosol composition of atmospheric suspended particles in our model. Therefore, we regard water as an aerosol species.

*L243: "2018, when rain events occurred". Where is this shown?*

Response:

Information about rain observation was added to Figure 1a and section 2.1. In addition, the sentence

was revised as presented below.

Before revision>> "North of the equator, nss-SO$_4$$^{2-}$ and NH$_4$$^+$ concentrations were especially high. These mass fractions were, respectively, 70–76% and 18–22%, except for data of 11 November 2018, when rain events occurred."

After revision>> "These mass fractions were, respectively, 70–76% and 18–22%, except for data of 11 November 2018, when nss-SO$_4$$^{2-}$ and NH$_4$$^+$ were less around rain event occurrences."

*L243-244: ""The values of nss-K+, also tended to be north of the equator. What does it mean?*

Response:

The sentence has been corrected to "The values of nss-K$^+$ also tended to be higher north of the equator".

*L314-315: "that they have hygroscopicity". What does it mean?*

Response:

The sentence has been corrected to "that they are water-soluble materials".

*L316-317: "are probably formed with secondary formation". What does it mean?*

Response:

The sentence and the preceding sentence were corrected as presented below.

Before revision>>"Such particles are probably formed with secondary formation or coagulation of sulfate with an Fe-containing particle."

After revision>> "Such sulfate on Fe-containing particles can be formed by condensation from sulfuric gaseous materials or coagulation of sulfate particles."

*L319: I think "under their deliquescence humidity" can be deleted.*
(Probably L329)

Response:

We had intended to mention state in a metastable humidity (i.e. from deliquescence humidity to

efflorescence humidity). Although the particle state can be determined as liquid in higher-than-deliquescence humidity and solid in efflorescence humidity, the state in a metastable humidity can be solid or liquid depending on the experience of humidity. However, the state of ammonium sulfate particles in the atmosphere can be inferred from the particle morphology on the sample and in situ humidity because the shape of solid ammonium sulfate changes under the metastable humidity, as explained in text. Therefore, the sentence is corrected as presented below.

Before revision>>"Particles composed mainly of ammonium sulfate can be present as solid or liquid under their deliquescence humidity, according to atmospheric humidity experience."
After revision>>"Particles composed mainly of ammonium sulfate can be present as solid or liquid in a metastable humidity condition between efflorescence and deliquescence humidities, according to their atmospheric humidity experiences."

*L362: "the Fe-containing particles mixed with both Ca and Mg were only one". What does it mean?*

Response:
The sentence was corrected as presented below.
After revision>> ", in our study, an Fe-containing particle in which both Ca and Mg were detected was only one…"

*L 379: Replace "Sects" with "Sections"*
*L492: "Fe/Pd" should be in italics.*

Response:
Those points were corrected in the revised manuscript according to the comments.

*Notification to the authors from Editor:*

*Please ensure that the colour schemes used in your maps and charts allow readers with colour vision deficiencies to correctly interpret your findings. Please check your figures using the Coblis – Color Blindness Simulator (https://www.color-blindness.com/coblis-color-blindness-simulator/) and revise the colour schemes accordingly.*

Response:

We checked all figures using the simulator. Because some colors in Figures 1, 6, 9 and 10 were too light for some color vision deficiency categories, we changed them. Mapping photographs (Figures 4 and 7) and Maps (Figure 2 and Figure 8) were also checked; all patterns of the simulator were checked. The colors were looked different, but same colors were not assigned to different values. The color contrast seemed sufficient to recognize important differences of values for this study in all cases. Therefore, we judged that our map colors can accommodate color vision deficiencies and can enable readers to interpret our findings correctly.

---

## Author Response (AR3)

*Editor Comments:*

*I thank the authors for significantly improving the readability of the manuscript and for addressing the highlighted points. However, I still have one final point that requires clarification.*

Response:

We appreciate your comment and terribly sorry that our errors remained in the manuscript.

*The concentrations shown in Tables 2 and 3 are now in ng/g; however, the concentrations from Figures 2 and 9 are in ng/kg. It seems that something is wrong because the Fe concentrations reported in Table 3 are 67 and 4.3 ng/g, but in Figure 9 a Fe concentration of 66.7 ng/kg is reported. Please verify the correct concentrations and use consistent units along the text.*

Response:

We corrected the units "ng/g" in Tables 2 and 3 to "ng/kg". In addition, values in Table 3 were united to a place of decimals. We also changed units and values of mass concentration in text of last paragraph in section 3.1 to uniform the unit from "ng m$^{-3}$" to "ng/kg".

---

## Author Response (AR4)

We changed a postal guide number and added some funds to acknowledgements. There are not the other changes.

P1

3 Japan Agency for Marine-Earth Science and Technology, Yokohama, 236-0001, Japan

P34

**Acknowledgments**

…..This study was supported by Ministry of Education, Culture, Sports, Science, and Technology and the Japan Society for the Promotion of Science (MEXT/JSPS) KAKENHI Grants 18H03369, 18J40204, 19K20438, 22K18023, JP19H04253, JP19H05699, JP19KK0265, JP20H00196, JP20H00638, JP21K12230, JP22H03722, JP22F22092, JP23H00515, JP23H00523, and JP23K18519, MEXT Arctic Challenge for Sustainability phase II (ArCS-II; JPMXD1420318865) projects, and the Environment Research and Technology Development Fund 2–2003 (JPMEERF20202003) and 2–2301 (JPMEERF20232001) of the Environmental Restoration and Conservation Agency. M.L. acknowledges the support of JSPS Postdoctoral Fellowships for Research in Japan (Standard).